# Macroscale and microcircuit dissociation of focal and generalized human epilepsies

Yifei Weng[1,2], Sara Larivière[2], Lorenzo Caciagli [3,4], Reinder Vos de Wael[2], Raúl Rodríguez-Cruces [2], Jessica Royer [2], Qiang Xu[1], Neda Bernasconi[2], Andrea Bernasconi[2], B. T. Thomas Yeo[5], Guangming Lu [1], Zhiqiang Zhang[1,6✉] & Boris C. Bernhardt[2,6✉]

Thalamo-cortical pathology plays key roles in both generalized and focal epilepsies, but there is little work directly comparing these syndromes at the level of whole-brain mechanisms. Using multimodal imaging, connectomics, and computational simulations, we examined thalamo-cortical and cortico-cortical signatures and underlying microcircuits in 96 genetic generalized (GE) and 107 temporal lobe epilepsy (TLE) patients, along with 65 healthy controls. Structural and functional network profiling highlighted extensive atrophy, microstructural disruptions and decreased thalamo-cortical connectivity in TLE, while GE showed only subtle structural anomalies paralleled by enhanced thalamo-cortical connectivity. Connectome-informed biophysical simulations indicated modest increases in subcortical drive contributing to cortical dynamics in GE, while TLE presented with reduced subcortical drive and imbalanced excitation–inhibition within limbic and somatomotor microcircuits. Multiple sensitivity analyses supported robustness. Our multiscale analyses differentiate human focal and generalized epilepsy at the systems-level, showing paradoxically more severe microcircuit and macroscale imbalances in the former.

[1] Department of Medical Imaging, Jinling Hospital, Nanjing University School of Medicine, Nanjing, China. [2] McConnell Brain Imaging Centre, Montreal Neurological Institute and Hospital, McGill University, 3801 University Street, Montreal, QC H3A2B4, Canada. [3] University College London Queen Square Institute of Neurology, London, United Kingdom. [4] Department of Bioengineering, University of Pennsylvania, Philadelphia, PA, USA. [5] Department of Electrical and Computer Engineering, Centre for Sleep and Cognition, Clinical Imaging Research Centre and N.1 Institute for Health, National University of Singapore, Singapore, Singapore. [6] These authors contributed equally: Zhiqiang Zhang, Boris C. Bernhardt. ✉email: zhangzq2001@126.com; boris.bernhardt@mcgill.ca

Widespread recurrent connections between thalamus and cortex represent a cardinal principle of brain organization, and thalamo-corticall interplay has long been recognized to contribute to whole-brain function and dysfunction[1,2]. As thalamo-cortical loops play a major role in the balance of cortical excitation and inhibition, a detailed assessment of perturbations in this network may enhance our understanding of mechanisms giving rise to the spectrum of common human epilepsies. A key involvement of the thalamo-cortical system is recognized in both genetic/idiopathic generalized epilepsies (GE), as well as syndromes traditionally considered, with temporal lobe epilepsy (TLE) as a prototypical example. Long-standing evidence from animal models and electro-clinical observations in patients indicates a prominent role of thalamo-cortical loops in seizures that appear generalized from the get-go, as in the case of GE. This is also seen in seizures propagating from a confined temporal network toward a more widespread hemispheric territory, with possible secondary generalization as in the case of TLE[3–5]. These observations are complemented by neuroimaging work in humans showing structural and functional changes in thalamic and cortical areas in both syndromes[6]. Collectively, these multiple lines of evidence suggest that investigating thalamo-cortical and cortico-cortical networks may be instrumental to understand mechanisms of focal and generalized human epilepsies at a systems level.

Despite early-day neuroimaging studies based on magnetic resonance imaging (MRI) volumetry of the thalamus in both TLE and GE[7], surprisingly little work directly addressed common and distinct perturbations of the thalamo-cortical network in both syndromes. Work comparing these cohorts in isolation to healthy controls (HCs) suggests that both generally present with morphological anomalies in this circuit, with findings being more robust in TLE than GE[6,8–12]. TLE indeed has been shown to present rather consistently with thalamic atrophy, specifically with marked effects in anterior and mediodorsal divisions, as well as widespread and multi-lobar cortical thinning[6,9], findings also supported by postmortem and ex vivo histological data[13,14]. GE, on the other hand, presents with a more mixed pattern of structural compromise, sometimes showing subtle atrophy[6,12,15] and sometimes not[16,17], possibly resulting from intrinsic heterogeneity across GE syndromes and a less severe disease trajectory. However, even when restricting GE patients to those with generalized tonic–clonic seizures as their only seizure type (i.e., not studying patients with juvenile absence or myoclonic epilepsies), findings remain inconsistent. Importantly, there are no studies directly comparing focal and GE in terms of thalamo-cortical and cortico-cortical network pathology, and underlying functional dynamics. Furthermore, neuroimaging findings have thus far not been related to potential microcircuit mechanisms that may play a critical role in shaping the macroscopic expression of epilepsy-related network abnormalities[18].

Advances in MRI acquisition and modeling techniques offer unprecedented opportunities to probe thalamo-cortical network organization and pathology across multiple scales in vivo[19]. In particular, it is now possible to interrogate different MRI contrasts and to aggregate those into regional descriptors of disease load at the morphological and microstructural level. Diffusion MRI tractography and resting-state functional MRI (rs-fMRI) connectivity analysis provide novel ways to profile brain organization at the network level[20,21]. Moreover, the synergistic integration of these different modalities addresses structure–function coupling and can be harnessed to provide large-scale simulations of brain function[22–25]. One approach specifically models whole-brain dynamics via a network of anatomically connected neural masses. In contrast to statistical approaches that interrogate macroscale organization and structure–function coupling, the

dynamical properties in these models are governed by parameters with biophysical interpretations that reflect empirical models of neuronal circuit function[22]. A recent study in healthy individuals has also shown that these models can make robust simulations of functional connectivity from structural connectivity, and that the model can furthermore be used to infer regional-specific microcircuit parameters, such as recurrent excitation–inhibition and external subcortical drive[22]. Thus, applying these models to epilepsy will provide a complementary and mechanistic perspective on the role of the thalamo-cortical system on cortico-cortical dynamics.

To study shared and distinct patterns of thalamo-cortical network pathology across the spectrum of focal and GE, we derived patient-specific measures of cortical and thalamic disease load using a multiscale MRI approach that targets microstructure, morphology, and macroscale connectivity. To identify microcircuit-level mechanisms dissociating both syndromes, we harnessed advanced biophysical simulation paradigms that integrate structural and functional connectome data in a unified framework, allowing the estimation of the role of recurrent excitation–inhibition as well as subcortical drive on cortical dynamics. To foreshadow our results, we observed more marked imaging anomalies in TLE relative to controls and GE, suggesting that this "focal" epilepsy is paradoxically associated with more severe whole-brain anomalies than an epilepsy syndrome defined by generalized seizures. Biophysical simulations complemented these findings and indicated marked reductions of subcortical drive together with increases in recurrent excitation–inhibition in TLE, specifically in limbic and fronto-central networks, while GE presented with increased subcortical drive but no changes in recurrent cortical excitation–inhibition. Several sensitivity analyses confirmed the robustness.

## Results

**Data sample and overall analytical strategy**. Our main analyses compared both patient cohorts directly to each other (see "Methods" section for inclusion criteria, clinical characteristics, and neuroimaging). In brief, we studied 107 TLE patients with unilateral hippocampal atrophy, 96 GE patients with generalized tonic–clonic seizures as their only seizure type, and 65 HCs. Patient cohorts had a comparable age and sex distribution, and underwent identical 3T multimodal MRI. Our image processing allowed profiling of cortical and thalamic morphology, microstructural markers as derived from diffusion MRI, and thalamo-cortical resting-state functional connectivity. Cortex-wide functional connectomes and diffusion tractography-derived structural connectomes were integrated using relaxed mean-field models. This computational technique was used to generate veridical simulations of functional connectivity from structural connectomes and to estimate cortical microcircuit parameters, specifically recurrent excitation–inhibition and influence of external/subcortical drive on macroscale cortical dynamics. The main findings show direct comparisons between patient cohorts for cortical (and cortico-cortical) as well as thalamic (and thalamo-cortical) networks, at the level of morphology, microstructure, connectivity, and microcircuit organization. Findings relative to the HCs are presented in the Supplementary Materials. All analyses controlled for age and sex.

**Morphological profiling**. Surface-based analysis of high-resolution MRI data in all participants profiled cortical and thalamic morphology (Fig. 1). These findings showed a marked divergence between GE and TLE, with the latter showing stronger atrophy in both cortical and thalamic regions. TLE patients furthermore presented with extensive cortical thinning in fronto-

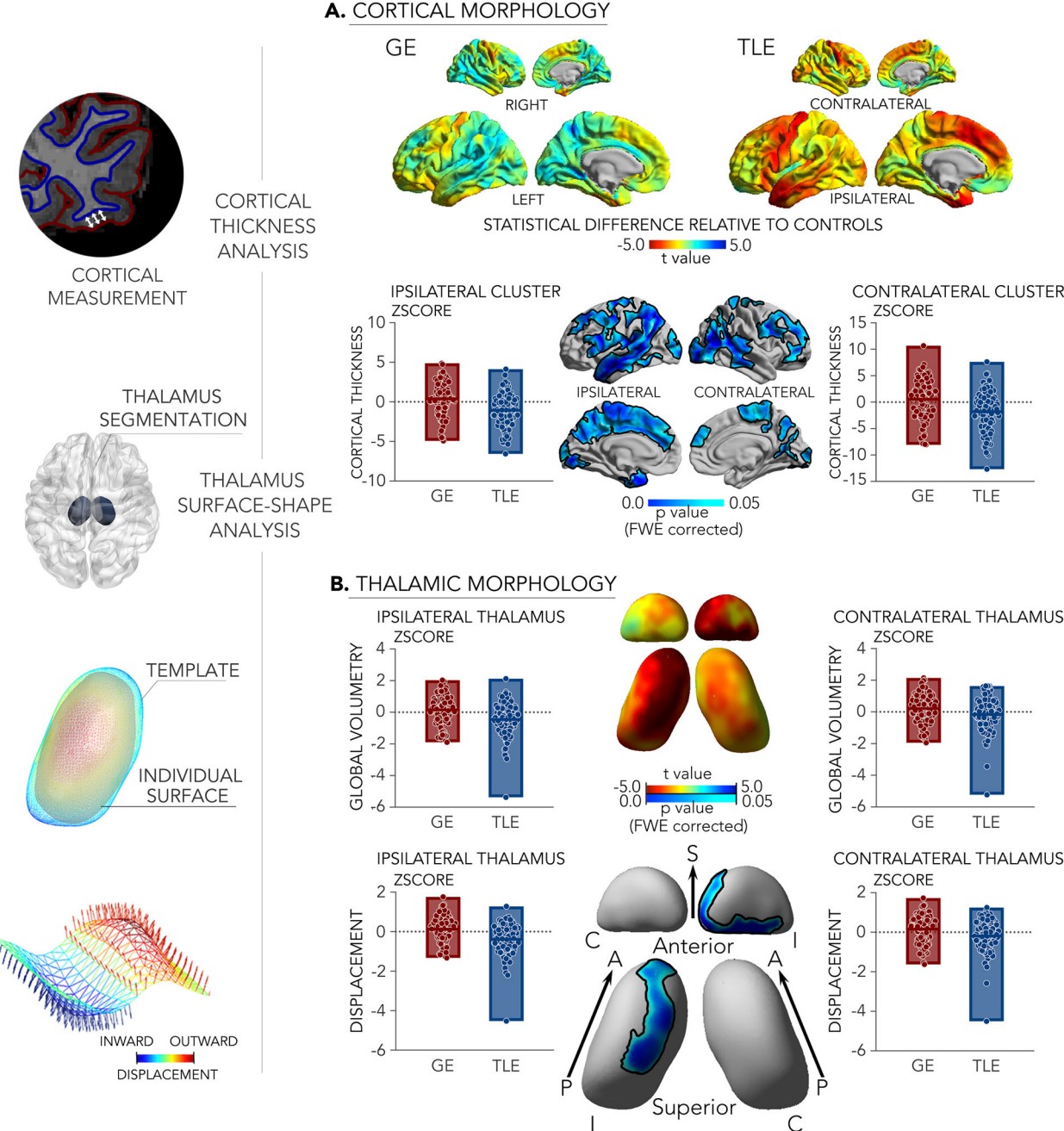

**Fig. 1 Morphological patterns were studied using a combined cortical thickness, thalamic volumetry, and thalamic surface-shape analysis (left).**
**a** Cortical morphological differences between 107 patients with TLE and 96 idiopathic/genetic GE. Surface-based findings were corrected for multiple comparisons at a family-wise error level of 0.05. **b** Thalamic morphological differences. Boxplots lines/edge denote mean/range. For comparisons against controls, see Supplementary Fig. 1a. S/A/P superior/anterior/posterior, C/I contralateral/ipsilateral.

central, temporo-limbic, prefrontal, and somatomotor regions (family-wise error corrected *p*-value, $p_{FWE} < 0.05$) relative to GE. TLE patients also showed bilateral thalamic volume reduction compared to GE (left and right $p < 0.005$). Surface-shape analysis confirmed strongest thalamic atrophy in ipsilateral mediodorsal divisions ($p_{FWE} < 0.05$). Similar findings were observed when separately comparing TLE and GE patients to HCs (Supplementary Fig. 1a), specifically showing marked atrophy in the former group and only subtle anomalies in the latter.

**Diffusion MRI studies of tissue microstructure.** Similar to the morphological findings, uni- and multivariate analyses of diffusion MRI parameters pointed toward a dissociation of GE and

TLE (Fig. 2). Indeed, surface-based diffusivity analysis of the superficial white matter, located immediately below the cortical mantle, revealed an extended territory of more marked anomalies in TLE compared to GE, encompassing temporal, fronto-opercular, as well as midline regions, with stronger effects in the ipsilateral hemisphere. Post hoc univariate analysis in clusters of multivariate findings, together with unconstrained surface mapping of individual diffusion parameters, showed that findings were generally characterized by decreases of fractional anisotropy (FA) and mean diffusivity (MD) increases in TLE ($p_{FWE} < 0.05$).

Complementing neocortical findings, multivariate diffusion parameter analysis of the thalamus identified marked differences between TLE and GE in both the left and right hemispheres.

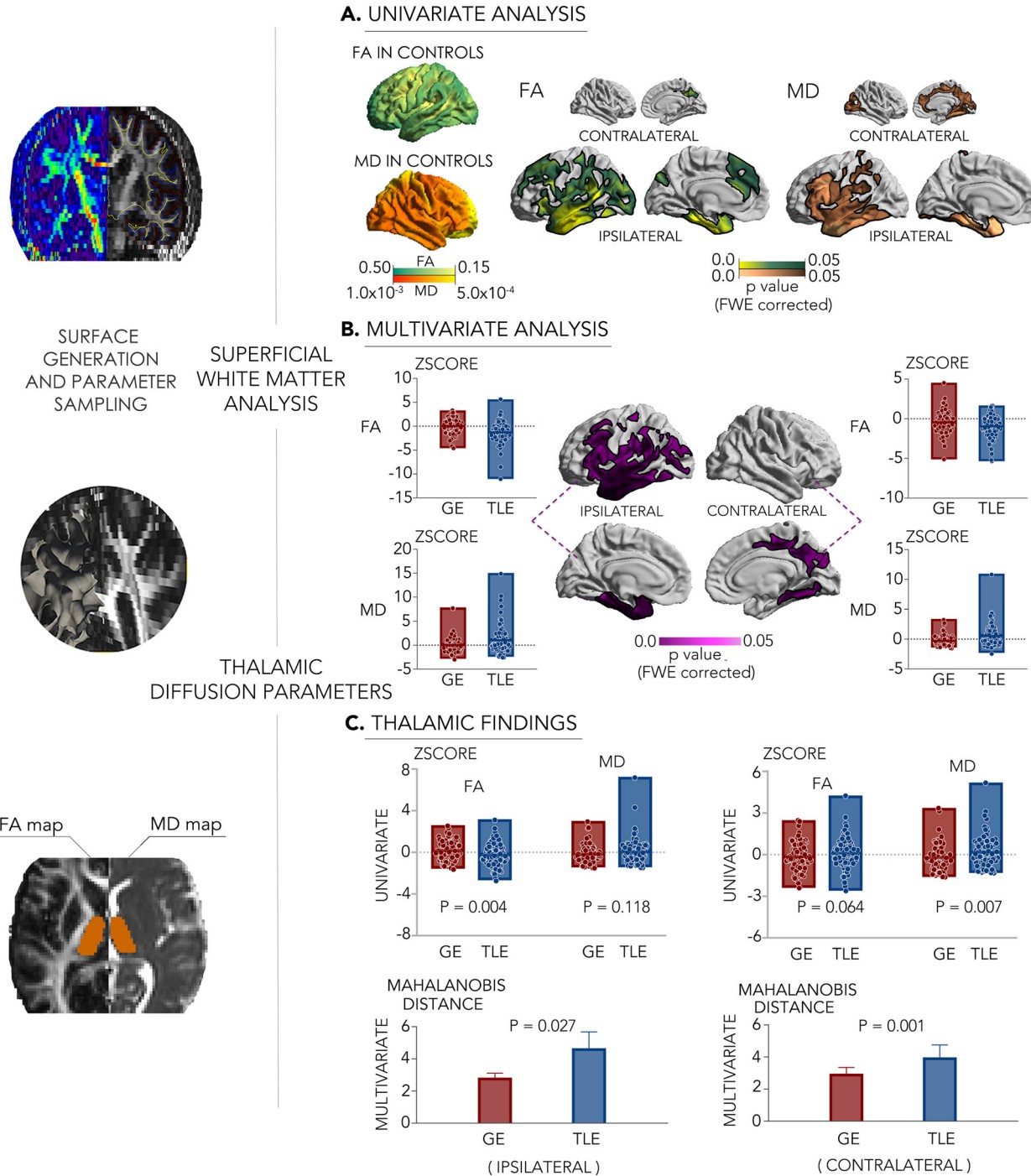

**Fig. 2 Superficial white matter and thalamic diffusion MRI profiling (left). a**, **b** Uni- and multivariate superficial white matter differences between 107 patients with TLE and 96 with GE. Surface-based findings were corrected for multiple comparisons at a family-wise level of 0.05. **c** Uni- and multivariate findings in the thalamus. Line and edge of boxplots denote the mean and range of the value. The error bar represents standard deviation. For comparisons of each cohort relative to controls, see Supplementary Fig. 1b.

Findings were characterized by marked FA reductions ($p < 0.005$) and marginal MD increases in the ipsilateral thalamus in TLE relative to GE. A similar dissociation between TLE and GE, again pointing toward more marked changes in the former cohort, was seen when comparing patient groups separately to controls (Supplementary Fig. 1b).

**Connectivity and microcircuit simulations**. We first built mean-field models to estimate the contributions of subcortical drive and recurrent excitation–inhibition on simulated cortico-cortical

dynamics, and then targeted thalamo-cortical functional networks using seed-based resting-state functional connectivity analysis (Fig. 3).

For biophysical circuit simulations, we leveraged a relaxed mean-field approach that models whole-brain functional dynamics and connectivity based on diffusion MRI connectome information (see "Methods" section for details). In brief, these models assume that neural dynamics are governed by (i) recurrent intraregional input related to recurrent excitation–inhibition, (ii) inter-regional input mediated by

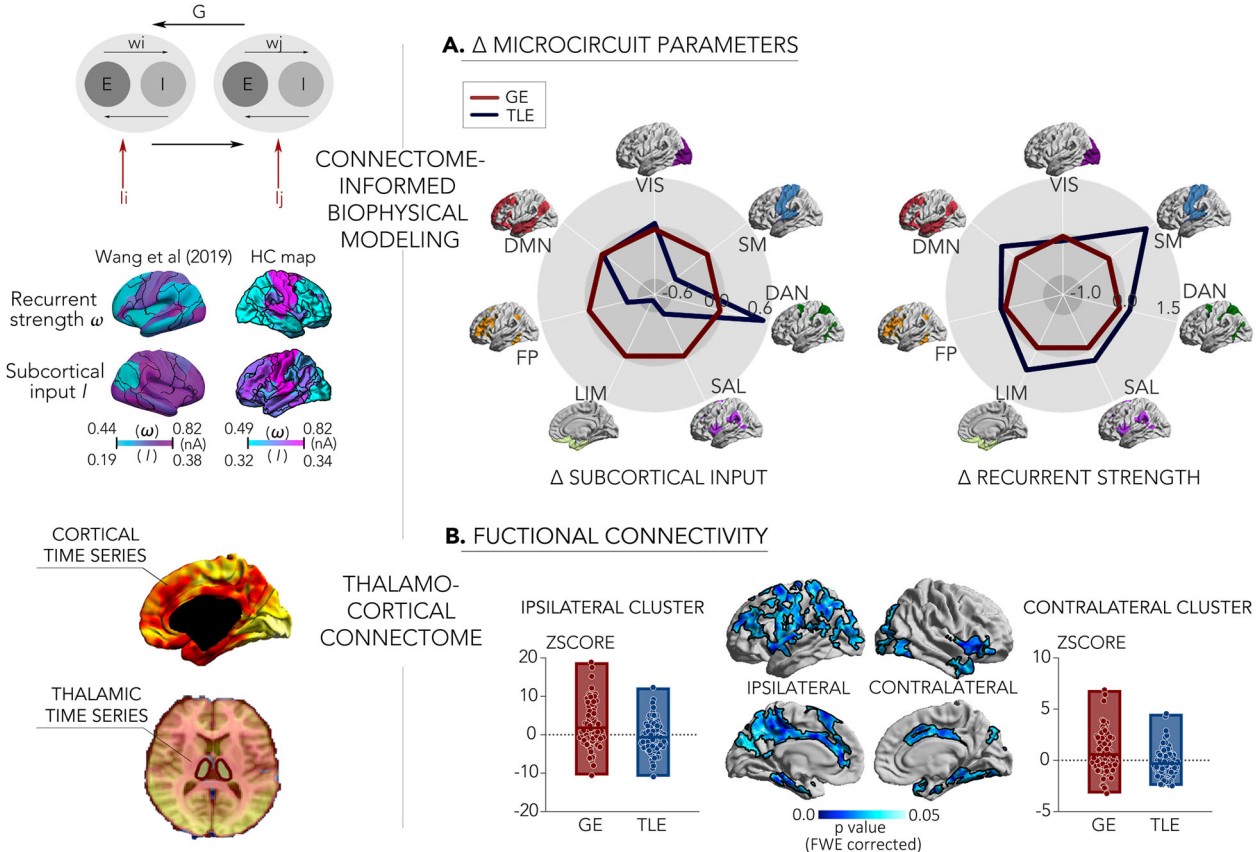

**Fig. 3 Functional and microcircuit divergences based on biophysical modeling and functional connectivity analysis (left).** Schema of the connectome-informed biophysical simulation, showing a similar distribution of model parameters in our 65 HCs compared to a previous study[22], and thalamo-cortical functional connectivity analysis (right). **a** Results from connectome-informed biophysical models, showing decreased subcortical drive but increased recurrent excitation–inhibition on cortico-cortical dynamics in 107 patients with TLE relative to 96 patients with GE. Black outlines indicate the boundaries of well-established intrinsic functional networks[86]. **b** Differences in thalamo-cortical functional connectivity between TLE and GE. Findings were corrected for multiple comparisons at a family-wise error level of 0.05. For comparisons of TLE and GE relative to HCs, see Supplementary Fig. 1c.

anatomical connections, (iii) extrinsic input mainly mediated by subcortical regions, and (iv) neuronal noise[22]. While the originally proposed mean-field models assume these four parameters to be constant across brain regions, the relaxed mean-field model applied here allows recurrent excitation–inhibition and subcortical input to vary, and to estimate these parameters at a region-specific level[22]. In our controls, the relaxed mean-field models showed a similar overall performance to simulated functional connectivity from structural connectivity, and yielded similar microcircuit parameter estimates as in a previous study in healthy adults[22] (Supplementary Fig. 2). For both patient cohorts, simulations supported the divergence between TLE and GE, demonstrating that cortico-cortical dynamics were driven by distinct microcircuit perturbations. Specifically, when compared to GE, TLE presented with reduced subcortical drive to multiple networks, predominantly to limbic and somatomotor networks. Conversely, recurrent excitation–inhibition to those two networks was increased in TLE relative to GE. Similar findings were observed when comparing each patient cohort separately to controls, mainly showing reduced subcortical drive in TLE and increased thalamo-cortical drive in GE (Supplementary Fig. 1c).

Considering thalamo-cortical functional connectivity, both patient groups diverged, with TLE showing widespread reductions in connectivity to nodes of the limbic, default-mode, fronto-parietal control, and somatomotor networks relative to GE ($p_{FWE}$

< 0.05). Similar findings were observed when independently comparing each patient cohort to controls, showing reduced thalamo-cortical connectivity in TLE and increases in GE (Supplementary Fig. 1c).

**Sensitivity analyses.** Several sensitivity analyses assessed robustness and consistency of our main findings. First, we repeated the thalamo-cortical functional connectivity analysis after additionally regressing out the global mean signal and observed virtually identical effects, i.e., a marked divergence between TLE and GE patients at the level of thalamo-cortical resting-state functional connectivity, with TLE showing markedly reduced connectivity relative to GE (Supplementary Fig. 3).

Second, we split the TLE cohort based on patients, who reported secondary generalized tonic–clonic seizures within a year prior to the imaging investigation (with/without secondary generalized tonic–clonic seizures: 31/69% of the TLE group), and repeated all elements of the multiscale analysis. This approach was chosen to reflect recent clinical history; patients with no or generalized seizure events in their history that were more than a year prior to the investigation were grouped into the 'without' category. Findings remained virtually identical when separately comparing each of these subcohorts to GE, suggesting that the dissociation between TLE and GE is not driven by current presence of secondary generalized tonic–clonic seizures in TLE, nor its absence (Supplementary Fig. 4).

Given previous work suggesting an association between neuroimaging measures of brain organization and drug response[16,26–29], we also repeated our analysis while additionally controlling for drug response patterns (i.e., drug resistance versus drug responsiveness). The linear model was set as followed: Model $= \beta_0 + \beta_1 \times$ Sex $+ \beta_2 \times$ Age $+ \beta_3 \times$ Medication response $+ \beta_4 \times$ Group. Focusing on relatively well-controlled (67 TLE, mean number of meds $= 1.27$; 83 GE, mean meds $= 1.16$) or refractory (40 TLE, mean meds $= 2.90$; 13 GE, mean meds $= 2.69$) patients, we still observed the above-mentioned major dissociations between patient groups, suggesting that divergent effects were likely independent from variable degrees of drug response in TLE and GE (Supplementary Fig. 5).

As both measures can affect brain structure and function, we repeated our analyses using statistical models that additionally controlled for age at onset and duration. These findings were highly consistent with our main results (Supplementary Fig. 6). Furthermore, findings were robust when restricting our cohorts to patients with at least 1-year epilepsy duration and at least one seizure per year (94 TLE and 74 GE) (Supplementary Fig. 7).

To finally investigate whether our results would be affected by hippocampal atrophy, we repeated the multiscale analysis after adding overall hippocampus volume as a covariate of no interest into the model. The linear model was thus set as follows: Model $= \beta_0 + \beta_1 \times$ Sex $+ \beta_2 \times$ Age $+ \beta_3 \times$ Mean hippocampal volume $+ \beta_4 \times$ Group. Although between-cohort differences were overall smaller due to dominant hippocampal lesions in TLE, main divergences remained (Supplementary Fig. 8).

## Discussion

Converging evidence from experimental studies in animals as well as electrophysiological and neuroimaging work in patients suggests a key role of the thalamo-cortical circuit in both focal and GE[3–6,30]. Yet, differential patterns of cortico-cortical and thalamo-cortical compromise across the spectrum of human epilepsies are not fully understood, due to the scarcity of work systematically and directly comparing focal and GE. To close this gap, we profiled thalamo-cortical network pathology in two large groups of TLE and GE patients, as well as matched controls, using multiscale neuroimaging and computational simulations of microcircuit function. Despite gathering evidence that the structural and functional organization of the thalamo-cortical network is abnormal in both patient groups relative to controls, we obtained robust support for a dissociation of both syndromes at the level of morphology, diffusion MRI-derived indices of tissue microstructure, and macroscale functional connectivity. Findings indicated more marked anomalies in TLE relative to controls and GE, thus suggesting that the archetypal "focal" epilepsy is paradoxically associated with more severe whole-brain anomalies than an epilepsy syndrome electro-clinically defined by generalized seizures. Further support for a dissociation between both syndromes was gathered from connectome-informed biophysical simulations. These suggested marked reductions of subcortical drive, as well as increases in recurrent excitation–inhibition on cortical microcircuit function in TLE, specifically in limbic and fronto-central networks, while GE presented with increased subcortical drive but no changes in recurrent cortical excitation–inhibition. Findings were not mediated by between-cohort differences drug response and remained significant when separately analyzing TLE cohorts according to a history of secondary generalized tonic–clonic seizures. Furthermore, although anomalies in TLE were in part diminished when controlling for hippocampal structural imaging findings, further confirming that mesiotemporal pathology often mirrors the degree of whole-brain anomalies in this syndrome[31], differences between TLE and GE were still largely consistent. Collectively, our findings demonstrate thalamo-cortical pathological signatures in TLE and GE that convey a compelling dissociation of the two syndromes at multiple scales, refining our understanding of macroscale and microcircuit mechanisms contributing to the spectrum of common epilepsies in humans.

Our work benefitted from inclusion of a large sample of patients with TLE and GE in whom high-quality 3T multimodal neuroimaging data were available, together with a matched sample of HCs. Our neuroimaging paradigm leveraged cutting-edge morphological analysis, diffusion parameter profiling, and macrolevel connectomics of both cortical and thalamic subregions. While prior research from our group and others has shown that the integration of these features can lead to novel descriptions of healthy and diseased brains[19,32], the current work represents one of the most comprehensive in vivo study to date of thalamo-cortical and cortico-cortical networks in both epileptic conditions. Combining these different imaging modalities, our findings strengthen the notion that both GE and TLE are associated with disorganization of these different networks at multiple scales[33–36]. However, and even more importantly, we captured consistent evidence for diverging patterns of pathology across both syndromes. Considering gray matter morphology, TLE patients presented with marked atrophy in dorsomedial thalamic divisions as well as prefrontal, fronto-central, and temporal neocortices relative to GE patients and relative to controls, paralleling earlier studies suggesting an association between TLE and multi-lobar gray matter reductions demonstrated, using several techniques and confirmed across multiple sites[6,37,38]. Our inclusion criteria ruled out the presence of visible neocortical lesions in the investigated patient cohorts and, overall, the pathological substrates of gray matter atrophy in TLE remain unclear. Yet, cortical atrophy appears relatively prevalent across TLE cohorts, with marked gray matter loss of temporal and fronto-central regions seen in ~35% of patients[39]. Although the presence of subtle preexisting anomalies of neurodevelopmental origin is possible[40], cortical atrophy undergoes measurable progressive changes over time[37,41–43]. These findings are also in line with the conclusions of a prior postmortem study in TLE patients, suggesting acquired cortical pathology that potentially represents seizure-related damage and manifests as both gliotic and microgliotic changes[14]. In the thalamus, while postmortem analysis also emphasized heterogeneity across patients, findings overall confirmed reductions in neuronal density in mediodorsal divisions, which could represent a plausible pathological substrate of the local volume reductions seen in the current work[13]. On the other hand, comparison of GE patients to controls highlighted gray matter loss only at uncorrected levels in fronto-central regions, reflecting a generally less marked impact of GE on mesoscale morphology[12,16,44]. In this work, gray matter morphometric analyses were complemented by diffusion MRI parametrization of the thalamus and the superficial white matter, which lies immediately below the cortical interface. This approach offered an analysis of diffusion MRI metrics in the same anatomical reference frame as the morphological findings. Although this approach detected diffusion anomalies in GE in both cortical and thalamic regions, findings in this cohort were more restricted than those observed in TLE. Indeed, diffusion anomalies in TLE affected the thalamus bilaterally, as well as a widespread cortical territory that radiated outward from the ipsilateral mesiotemporal disease epicenter to invade limbic and higher-order association cortices in temporal frontal and parietal lobes. As the diffusion parameter changes identified in this work may be sensitive to different microstructural and architectural substrates, we can only speculate on histopathological underpinnings. In TLE, prior work has shown that FA reductions of select fiber tracts may relate to

changes in axonal membrane and myelination[45], whereas MD changes may represent a combination of gliosis and increased extracellular space[46,47]. Collectively, the analysis of TLE and GE indicates that both syndromes affect thalamic and cortical networks. On the other hand, our findings strongly suggest diverging pathological signatures at the macroscale level, contributing to our increasing understanding of whole-brain disease effects across the common epilepsies.

Given the maturation of MRI co-registration techniques to bring different modalities into the same reference frame[48], studies have begun to investigate the coupling between brain structure and function in humans[49]. Our work harnessed biophysical simulations to integrate structural and functional network data, and to estimate aspects of cortical microcircuit organization, specifically the interplay of recurrent excitation–inhibition and the influence of subcortical/external drive on cortical dynamics. The relaxed mean-field models chosen here provide plausible estimates of functional connectivity from structural connectomes with relatively few assumptions and low parametric complexity. In healthy populations, these techniques have shown promise to simulate functional interactions solely based on structural connectivity with robust accuracy, and have begun to provide insight into the hierarchical organization of cortex at macroscale and its link to underlying microcircuit configurations[22–25,50]. Capitalizing on one of the first applications of these relaxed mean-field models to epilepsy cohorts, we identified diverging mechanisms in focal and GE, bridging microcircuit properties and macroscale function. In fact, these large-scale models allowed for the estimation of region-specific model parameters that provided insights into the role of excitation–inhibition and excitatory subcortical drive on microcircuit function. These analyses suggested marked divergences between epilepsy syndromes with respect to the role of subcortical input on cortico-cortical dynamics, with TLE showing a reduced influence, particularly to limbic and somatomotor areas, while GE expressed an increasing contribution of subcortical drive on cortical function. The increased subcortical drive in GE did not seem to be accompanied by marked changes in recurrent excitation–inhibition in cortical regions, a finding that may mirror the overall smaller degree of morphological and microstructural anomalies observed in our analyses. Despite unchanged intraregional recurrent input, cortical circuits in GE may still show greater excitability due to increased subcortical drive. Conversely, reductions in subcortical drive in limbic and somatomotor areas in TLE occurred in parallel with increased recurrent excitation–inhibition, echoing the stronger morphological and microstructural anomalies detected in fronto-central cortices in this cohort.

According to prior neuroimaging studies in patients and experimental work performed in animal models, persistent and often recurrent loops between the thalamus and distributed neocortices in GE may contribute to the initiation and maintenance of generalized seizures[51]. In contrast, thalamo-cortical decoupling has previously been detected in TLE via rs-fMRI[4], structural MRI covariance[9], and diffusion MRI tractography[52]. The overall disconnection of limbic cortices may possibly relate to a functional segregation of these regions from subcortical networks, which in turn may reflect the existence of recurrent cortico-cortical loops in TLE, and potentially promote epileptogenesis. In future work, it will be relevant to further cross-validate the connectome-derived biophysical parameters with empirical studies on microcircuit function, and dysfunction at laminar and cellular scales[18], for example, through ultra-high field MRI studies or studies in animal models. Perturbations in thalamic connectivity to cortical target regions may, for instance, alter the microstructural context and connectivity between different cortical layers, in turn perturbing the cortical microcircuit layout and hierarchical organization of cortical networks[53]. Beyond contributing to efforts aiming at dissociating between established electro-clinically defined epilepsy syndromes, shifting toward an intracortical- and microcircuit-level scale may identify mechanisms contributing to inter-individual variability within particular syndromes, for example, by showing how seizures may generalize in a given patient and how seizure generalization could be prevented. By consolidating connectivity, structure, and microcircuit properties into a unified framework, these investigations could open new avenues for the conceptualization, and dissociation of focal and generalized human epilepsies.

Our results support the notion that both GE and TLE are associated with thalamo-cortical and cortico-cortical network abnormalities at multiple levels. Moreover, our findings demonstrated a robust dissociation between TLE and GE, with the former being more markedly affected. Despite this divergence, one cortical network that appeared most susceptible to epilepsy-related pathology overall was the limbic system, showing measurable microstructural anomalies in both focal and generalized syndromes. Increased susceptibility of the limbic system has repeatedly been suggested in the context of TLE[54,55], and is also plausible given the mesiotemporal epicenter of the disorder. In different GE syndromes, several studies have also detected structural anomalies in limbic cortices, including work showing mesiotemporal gray matter loss as well as malrotation[56–58]. The increased disease effect on the limbic system across both TLE and GE may relate to the role of the thalamus itself in the pathophysiological networks of both syndromes. Several divisions of the thalamus, such as the mediodorsal and anterior division, have sometimes also been closely associated to the limbic circuitry due to their anatomical proximity and dense interconnectivity with several key nodes of the "grand lobe limbique," such as the hippocampus and amygdala. On the other hand, limbic and paralimbic cortices differ from other cortices, in terms of their underlying microstructure and circuit properties[59,60], and have a relatively simple laminar organization compared to eulaminate areas exhibiting a more differentiated lamination[53,59,61]. These differential lamination patterns also mirror gradual changes in myeloarchitecture and microcircuit properties, with limbic areas showing less myelination but higher synaptic density compared to areas involved in less integrative sensory–motor and unimodal association processing. The latter are ultimately believed to affect the increased plasticity and heightened susceptibility of the limbic network to multiple diseases[59].

Our inclusion criteria restricted the analysis to patients with an electrophysiological signature characteristic of TLE, as well as GE patients with tonic–clonic seizures as their only seizure type. Hence, despite the consistent dissociation of these two prototypical forms of focal and GE shown in this work, it remains to be evaluated whether and how our framework extends to other prevalent generalized and focal epilepsy syndromes, including extratemporal epilepsy secondary to malformations of cortical development, that is associated with an intriguing, and seemingly coupled, spectrum of lesional pathology, and widespread network anomalies[62]. On the other hand, our study benefitted from the inclusion of patients with different disease severities, clinical history, and hippocampal imaging findings, allowing us to examine the contribution of these inter-individual factors on our results. A comprehensive battery of sensitivity analyses revealed that between-cohort divergences were only modestly related to potential variations in drug response patterns, virtually identical when restricting the TLE cohort to those with a history of secondary generalized tonic–clonic seizures, and robust against correction for age at seizure onset and disease duration. Furthermore, although we found a noticeable modulation of the

between-cohort difference when factoring in the degree of hippocampal anomalies, a key indicator of network level compromise in TLE (refs. [31,63]), between-group divergences were still consistent.

We conclude by emphasizing that our work offers an integrative perspective on how different scales of brain organization interact in focal and generalized epilepsies. While our cross-sectional findings preclude direct inferences on whether microcircuit perturbations cause macroscale reorganization (or vice versa), they motivate future longitudinal research that models these multiscale interactions in the context of disease progression. Given their scope to interrogate microcircuit and macroscale effects, we will similarly explore the utility of the developed platform in the prediction, and predict and monitoring of the efficacy of therapeutic interventions, notably anti-epileptic medication and epilepsy surgery.

## Methods

**Participants**. We studied 263 epilepsy patients recruited from Jinling Hospital, Nanjing, China between July 2009 and August 2018. Patients were diagnosed as having either GE with generalized tonic–clonic seizures, or unilateral TLE with MRI evidence for hippocampal sclerosis. Diagnoses followed ILAE criteria[64], and were informed by electro-clinical factors, neurological examination, and neuroimaging. Further inclusion criteria were: (i) age older than 16 years; (ii) right-handedness; (iii) no mass lesion (i.e., brain tumor, cerebral hemorrhage or ischemia, and cerebrovascular malformation); (iv) no history of brain surgery; (v) no significant physical conditions; (vi) no alcohol or substance abuse; and (vii) no MRI contraindications. Among the initial 263 patients, we selected only those with available MRI data for all the studied modalities and those who did not present with imaging artifacts. Our final patient cohort consisted of 203 patients: 96 GE (31 females, mean ± SD age = 25.65 ± 7.85 years) and 107 TLE patients (53 left and 54 right TLE; 47 females, mean ± SD age = 27.29 ± 7.81 years).

Patients were compared to 65 age- and sex-matched HCs (25 females, mean ± SD age = 24.98 ± 4.96 years). Detailed sociodemographic and clinical information can be found in Table 1. This study was carried out according to the declaration of Helsinki and approved by the ethics committee of Jinling Hospital. Written informed consent was obtained from every participant.

**MRI protocol**. Data were acquired on a 3T MRI scanner (TIM Trio, Siemens Medical Solution, Erlangen, Germany) equipped with an eight-channel head coil. We used a 3D rapid gradient echo sequence to acquire high-resolution T1-weighted MRI (T1w; 176 slices; repetition time (TR) = 2300 ms; echo time (TE) = 2.98 ms; flip angle = 9°; field of view (FOV) = $256 \times 256$ mm$^2$; $0.5 \times 0.5 \times 1$ mm$^3$ voxels) and a 2D echo-planar imaging spin-echo sequence to acquire diffusion MRI (DWI; 45 slices; TR = 6100 ms; TE = 93 ms; 120 volumes with non-collinear directions (b = 1000 s/mm$^2$) and four volumes without diffusion weighting (b = 0 s/mm$^2$); FOV = $240 \times 240$ mm$^2$; $0.94 \times 0.94 \times 3$ mm$^3$ voxels). Using 2D echo-planar BOLD imaging, we acquired rs-fMRI (30 slices; TR = 2000 ms; TE = 30 ms; flip angle, 90°; FOV = $240 \times 240$ mm$^2$; 250 volumes; $3.75 \times 3.75 \times 4$ mm$^3$ voxels). Participants were instructed to keep their eyes closed and remain still in the

scanner. Two experienced radiologists (Z.Z. and G.L.) reviewed routine T1w and FLAIR images and reached consensus in the final diagnosis.

**Structural MRI**. We processed T1w data using FreeSurfer (v6.0; http://surfer.nmr.mgh.harvard.edu/)[65–67] to generate models of the cortical surface and to index neocortical morphology. In brief, the pipeline involves skull stripping, image registration, and cortical surface reconstruction. Cortical thickness was measured as the Euclidean distance between corresponding vertices on pial–gray and gray–white matter interfaces. Surfaces were inspected for inaccuracies, and corrected if necessary, prior to registration to the hemisphere-matched Conte69 template from the human connectome project initiative[68] with ~64 k surface points (henceforth, vertices). Thickness data were blurred using a 20 mm full-width at half-maximum kernel.

The entire thalamus was automatically segmented using FSL-FIRST (v5.0.9; https://fsl.fmrib.ox.ac.uk/fsl/fslwiki/FIRST/)[69], and segmentations were linearly registered to MNI152 space. Segmentations were converted to triangular surfaces and parameterized using a spherical harmonic representation model[70–72]. We generated a thalamic template with ~6k vertices across our HCs, and aligned individual surfaces to the template based on shape intrinsic features. We measured vertex-wise displacement vectors for each participant's thalamus to the template in surface normal direction, indicating inward/outward deformations of an individual relative to HCs[9].

**Diffusion MRI**. Data were preprocessed with MRtrix (v0.3.15; http://www.mrtrix.org/)[73]. Processing included head motion and eddy current correction, de-noising, as well as diffusion parameter estimation (FA, MD). As in previous analyses[32,74,75], we co-registered the T1w to diffusion MRI data using boundary-based registration[48]. To study the microstructure of the white matter immediately beneath the neocortical mantle, we generated subject-specific surfaces by propagating the gray–white matter interface along a Laplacian potential field toward the ventricular walls for approximately 2 mm[32,74,75]. This depth was selected to target the cortico-cortical U-fiber system as well as terminations of long-range fiber tracts[76]. Superficial white matter surfaces were mapped to diffusion space via the inverse to the initial co-registration, and used to sample voxel-wise FA and MD. As for the cortical thickness measures, superficial white matter data were surface-registered to Conte69.

To analyze thalamic diffusion parameters, preprocessed diffusion MRI data were mapped to the MNI152 template using a combination of linear and nonlinear transformations. Left and right thalamic masks were generated by intersecting the previously generated thalamic segmentations across participants in MNI152 space. We extracted thalamic FA and MD values for all participants prior to statistical analysis.

Preprocessed diffusion MRI data in native space underwent multi-shell, multi-tissue constrained spherical-deconvolution to estimate voxel-wise fiber orientation distributions. Anatomically constrained tractography generated 40 million tracts[77], and we stored structural connectomes of 200 cortical parcels, defined via subclustering of an anatomy-based atlas while guaranteeing comparable parcel size[78,79]. The group-averaged structural connectome was fed into the biophysical modeling framework (see below).

**Resting-state fMRI**. The rs-fMRI processing was conducted via DPARSF (v2.3; http://www.rfmri.org/DPARSF)[80]. The first 10 images were excluded to ensure steady-state signal equilibrium. Images underwent correction for slice timing, realignment, band-pass filtering (0.01–0.1 Hz), and spatial smoothing using a 6 mm full-width at half-maximum Gaussian kernel. We statistically corrected for head motion as well as average white matter and cerebrospinal fluid signals. To extract cortical functional time series, we aligned subject-specific functional images to their cortical surfaces via boundary-based registrations[22] and sampled time series at each vertex at mid-thickness. As with the cortical thickness measures, functional time series were also aligned to Conte69.

To obtain thalamic time series, we resampled native space rs-fMRI data to the MNI152 template. We used the above-mentioned thalamic masks to extract time series for all participants, and calculated thalamo-cortical functional connectivity via Pearson correlation coefficients between the thalamus and each cortical vertex. A cortex-wide functional connectivity matrix was also calculated based on the same cortical parcellation as above. Individual connectivity maps underwent Fisher $r$-to-$z$ transformations prior to generating a group-level connectome.

**Connectome-informed biophysical simulations**. We used a recently proposed connectome-informed brain model to study structure–function coupling and to estimate the relative influence of microcircuit parameters on macroscale cortical dynamics in our three cohorts[22]. Specifically, we harnessed a relaxed mean-field neural mass model that captures the link between cortical functional dynamics and structural connectivity derived from diffusion imaging, and its modulation through region-specific microcircuit parameters. For further details on the model and its mathematical underpinnings, we refer to the original publication on the relaxed mean-field model[22] and earlier work on the use of (non-relaxed) mean-field models in the context of connectomics[81].

---

**Table 1 Demographic data and clinical information. Age, onset, duration, and seizure frequency are presented in mean ± SD (range).**

|  | GE-GTCS | TLE-HS | HC |
|---|---|---|---|
|  | (n = 96) | (n = 107, L/R = 53/54) | (n = 65) |
| Sex (M/F) | 65/31 | 60/47 | 40/25 |
| Age (years) | 25.65 ± 7.85 (16–50) | 27.29 ± 7.81 (16–48) | 24.98 ± 4.96 (21–40) |
| Onset (years) | 19.72 ± 8.86 (7–49) | 16.28 ± 9.15 (0.25–43) | — |
| Duration (years) | 5.93 ± 6.36 (0.01–43) | 10.87 ± 8.76 (0.08–32) | — |
| Seizure frequency (per month) | 2.82 ± 9.86 (0.01–62) | 9.69 ± 22.69 (0.03–150) | — |
| Drug resistant (%) | 13 (13.5%) | 40 (37.4%) | — |

*L LTLE-HS, R RTLE-HS, M male, F female.*

In brief, mean-field neural mass models capture neural dynamics at the level of neuronal populations. This is achieved by the mathematical simplification (i.e., mean-field reduction) of detailed spiking neuronal networks. While making some simplifying approximations[81], mean-field models have been shown to capture complex neural dynamics with relatively low parametric complexity[82]. More specifically, under the mean-field model, the neural dynamics of a given region are governed by four components: (i) recurrent intraregional input, i.e., recurrent excitation–inhibition; (ii) inter-regional input, mediated by anatomical connections from other nodes, (iii) external input, mainly from subcortical regions, and (iv) neuronal noise modeled through an uncorrelated Gaussian[22]. There are "free" parameters associated with each component. The original (non-relaxed) mean-field models[81] assume these parameters are constant across brain regions, which is not biologically plausible. The relaxed mean-field model that we utilized allows recurrent excitation–inhibition, and extrinsic (subcortical) input to vary across regions and automatically estimated[22]. The dynamics of each region $i$ is described by the following coupled nonlinear stochastic differential equations[81]:

$$\dot{S}_i = -\frac{S_i}{\tau s} + r(1 - S_i)H(x_i) + \sigma \nu_i(t), \tag{1}$$

$$H(x_i) = \frac{ax_i - b}{1 - \exp(-d(ax_i - b))}, \tag{2}$$

$$x_i = wJS_i + GJ \sum_j C_{ij} S_i + I. \tag{3}$$

For a given region $i$, $S_i$ in formula (1) represents the average synaptic gating variable, $H(x_i)$ in formula (2) denotes the population firing rate, and $x_i$ in formula (3) denotes the total input current. In (3), the input current is determined by the recurrent connection strength $w_i$ (henceforth excitation–inhibition), the excitatory external input $I$ (such as from subcortical relays; henceforth subcortical input), and inter-regional signal flow. The latter is governed by $C_{ij}$, which represents the structural connectivity derived from diffusion MRI tractography between regions $i$ and $j$, and the global coupling $G$. The constant $G$ scales the strength of information flow from other cortical regions to the i-th region relative to the recurrent connection and subcortical inputs. Following prior work[22], we set synaptic coupling $J = 0.2609$ nA. In formula (2), parameter values for the input–output function $H(x_i)$ were set to $a = 270$ n/C, $b = 108$ Hz, and $d = 0.154$ s.

To run the model in each of the cohorts (GE, TLE, and controls), we entered the group-average structural and functional connectivity matrices as input, yielding the global coupling constant $G$, the global noise amplitude $\sigma$, as well as recurrent connection strengths $w_i$, and excitatory subcortical inputs $I_i$ for each region as output. During parameter estimation, simulated neural activities $S_i$ in formula (1) were fed to a Balloon–Windkessel hemodynamic model[83] to simulate fMRI signals for each region $i$. Global and region-specific parameters were determined by maximizing the similarity between simulated and empirical functional connectivity, based on a previously developed algorithm for inverting neural mass models that leverages the well-established expectation–maximization algorithm. Correlations between simulated and empirical functional connectivity were $r = 0.51/0.48/0.57$ in HC/GE/TLE (and above baseline correlations between structural and functional connectivity $r = 0.37/0.40/0.41$; Supplementary Fig. 2). To measure alterations in subcortical input on cortico-cortical dynamics, we quantified between-group differences for given region $i$ and normalized results by the SD within the corresponding network.

**Statistics and reproducibility**. (a) Surface-based analyses: Analysis was carried out using SurfStat for Matlab[84], which provides a unified framework to assess between-group differences in our multiscale measures (i.e., cortical thickness, thalamic surface-shape displacements, superficial white matter diffusivity, and thalamo-cortical functional connectivity), while controlling for age and sex. Surface-based measures were z-scored relative to HCs, and left and right TLE patients were pooled into a single cohort such that all lesions were consistently located in the left hemisphere[85] (findings were also similar when flipping a comparable proportion of GE patients; Supplementary Fig. 9). Surface-based results were corrected for multiple comparisons at a family-wise error rate of $p_{FWE} < 0.05$.

(b) Thalamic analysis: To study univariate thalamic parameters (i.e., global thalamic volume, FA, and MD), we z-scored the parameters relative to controls and measured group differences using Student's t-test. Mahalanobis distances were calculated as multivariate dissimilarity measures for comparing diffusion parameters in patients relative to controls, and significances were obtained using SurfStat's multivariate Hotelling's T2 test.

(c) Sensitivity analyses: Several sensitivity analyses assessed robustness and consistency of our main findings: (i) we repeated the thalamo-cortical functional connectivity analysis after additionally regressing out the global mean signal; (ii) we split the TLE cohort based on patients who reported secondary generalized tonic–clonic seizures within a year prior to the imaging investigation (with/without secondary generalized tonic–clonic seizures: 31/69% of the TLE group) and repeated all elements of the multiscale analysis and (iii) we repeated our analysis while additionally controlling for drug response patterns, age at onset and duration, and hippocampal volume.

**Reporting summary**. Further information on research design is available in the Nature Research Reporting Summary linked to this article.

**Data availability**

Surface feature data for all participants are available via osf.io (https://doi.org/10.17605/OSF.IO/GQXES).

**Code availability**

Statistical procedures from the SurfStat toolbox, together with documentation, are openly accessible via http://www.math.mcgill.ca/keith/surfstat and http://mica-mni.github.io/surfstat. Biophysical models can be accessed via https://github.com/ThomasYeoLab/CBIG.

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

## Acknowledgements

Y.W. acknowledges funding from China Scholarship Council (CSC: 201806190190). S.L. acknowledges funding from Fonds de la Recherche du Québec–Santé (FRQ-S) and the Canadian Institutes of Health Research (CIHR). R.V.d.W. receives support from a Savoy Foundation studentship). LC acknowledges support from Brain Research UK, and from a Berkeley Fellowship awarded by University College London and Gonville and Caius College, Cambridge. A.B. and N.B. were supported by FRQ-S and CIHR (MOP-57840, MOP-123520). B.T.T.Y. is supported by a Singapore National Research Foundation Fellowship (Class of 2017). Z.Z. acknowledges research funding from the National Science Foundation of China (NSFC: 81790653, 81790650, 81871345; 863 project: 2015AA020505), the National Key Technology (R&D) Program of the Ministry of Science and Technology (2018YFA0701703, 2017YFC0108805), China Postdoctoral Science Foundation (2016M603064), the Key Talent Project in Jiangsu Province (ZDRCA2016093), and Natural Scientific foundation-social development (BE2016751). B.C.B. acknowledges research funding from the SickKids Foundation (NI17-039), the National Sciences and Engineering Research Council of Canada (NSERC; Discovery-1304413), CIHR (FDN-154298), Azrieli Center for Autism Research (ACAR), an MNI-Cambridge collaboration grant, and the Canada Research Chairs Program.

## Author contributions

Y.W. designed the study, carried out the image processing and data analysis, interpreted the findings, and wrote and revised the manuscript. B.C.B. and Z.Z. designed the study, interpreted the findings, and wrote and revised the manuscript. Y.W., Z.Z., G.L., and Q.X. contributed to data collection and cohort selection. S.L. and R.V.d.W. assisted in image processing and data analysis, and revised the manuscript. L.C., N.B., A.B., B.T.T.Y., R.R.-C., and J.R. assisted interpretation of findings and revised the manuscript.

## Competing interests

The authors declare no competing interests.
