## [Peer Review File · Communications Biology]

Reviewers' comments:

Reviewer #1 (Remarks to the Author):

The manuscript "Macroscale and microcircuit dissociation of focal and generalised human epilepsies" presented multi-scale analyses, such as multimodal imaging, connectomics and computational simulations to dissociate human focal (TLE) and generalised epilepsy (GE) on the thalami-cortical pathology, especially the prominent role of thalami-cortical loops.

The authors find that TLE shows more severe microcircuit and macro scale imbalances, e.g., stronger cortical and thalamic atrophy, more bilateral volume reduction of thalamus, as well as widespread reductions in connectivity and reduced subcortical drive to the limbic and somatomotor networks. These new and solid evidences endow stronger support to dissociate human focal and generalised epilepsy at both the macroscale and microcircuit levels. The experimental findings and computational simulations are solid and significant, and the results are interesting and significant for both theoretical and experimental researchers in the field of epilepsy, especially for people who are trying to understand the underlying biophysical mechanisms of the seizure generalisation from the focal seizures.

However, I am still a little confused by the motivations and the discussions in the work have not touched this topic: The authors only presented results to dissociate TLE and GE, but did not try to clarify the non-consensus on the similarity or divergence on the role of thalami-cortical pathology in both generalised and focal epilepsies. As shown in the results of the manuscript, the difference between TLE and GE is so significant, even considering the comparison between TLE with secondary generalisation and GE. I am puzzled why there was nonconsensus and whether and how this work can clarify the nonconsensus. If so, the work can evoke more discussions on this problem and inspire further study. In my view, this part should be clearly introduced, presented and discussed in details. In this way, this manuscript will be more instrumental to understand mechanisms of focal and generalised human epilepsies, and thus contribute much more to this field.

Besides, the manuscript needs careful re-editing to clear the typos, grammatical mistakes as well as to carefully arrange the positions of the abbreviations.

Reviewer #2 (Remarks to the Author):

This paper presents an interesting connectivity study that compares the brains of people with generalized and focal epilepsies and compares these with controls. The study is well thought out, structured and reported. It presents novel results that could impact upon future understanding of these conditions and their treatment. Each aspect of the study provides useful and different insights that come together into a coherent story.

One of the key results is that there are "more marked anomalies in TLE relative to controls and GE". However, it is not clear how much this might related to lesions in the cortices of those with TLE. The selection criteria were no mass lesion and hippocampal lesions were considered, could individual's morphological and connectivity profiles be influenced by not-so-major cortical lesions that would come up as a broader effect in the combined results?

I am concerned about some of the details of the patients as presented in Table 1, and what impact the extremes might have upon the results. For instance, some patients have durations of 0.01 and 0.08

years, and some have seizure frequencies of 0.01 and 0.03 per month. These seem rather extremely small and further justification of inclusion of these patients is needed.

The model description in the biophysical simulations would benefit from increased explanations of the different components of the model for those who are less familiar with the models. Brief explanations of the forms of equations 1-3 would be useful. The model is stated to implement a "spiking neuronal network" but there is no spiking mechanism but rather a spiking rate function – an explanation of this would be helpful.

Minor:

- In the first sentence of the Introduction, clarify what "flexible" means.
- The figures contain a lot of text that is really small and difficult to read on the printed page. I suggest increasing the font size of these smallest fonts.
- I would like to know that the assumptions of the statistical tests were correctly considered.
- The proliferation of abbreviations is a little tiresome; I suggest keeping the abbreviations to only those terms used most often.

Reviewer #3 (Remarks to the Author):

Summary: The chief novelty of this study involves utilizing a relaxed mean field model as a computational technique to estimate linkages between structural and functional connectome measures, namely micro-circuit parameters produced from resting state and diffusion, along with macro-circuit features from brain volumetrics and surface shape analyses. The specific microcircuit parameters, whose calculation is not described, involve recurrent excitation/inhibition, in addition to external subcortical drive. These allow the authors to generate veridical simulations of functional connectivity from a structural connectome (diffusion based) that can be related linked to intrinsic functional connectivity networks and macrostructural cortical and thalamic morphology features of shape and volume. They contrast generalized epilepsy (GE) and temporal lobe epilepsy (TLE), with a focus on thalamo-cortical relationships and connectivity. The results clearly showed that the putatively more focal disorder of TLE is associated with widespread atrophy and surface-shaped changes in gray and white matter involving the thalamus and broader brain regions, a somewhat surprising finding given the broader electrophysiologic brain effects of GE. The relaxed mean field model technique allowed the investigators to examine measures related to subcortical drive and recurrent excitation/inhibition of cortico-cortico dynamics, as well as thalamic-cortical dynamics, and, again, TLE displayed more abnormalities than GE and in comparison to normal controls. More specifically, compared to GE, TLE showed decreases in subcortical drive that appeared related to increases in excitation/inhibition, particularly in limbic and fronto-central networks. In turn, this pattern in TLE demonstrated a greater association with multiple cortical intrinsic networks. In contrast, GE presented with increased subcortical drive, but no change in recurrent cortical excitation inhibition. The interpretation of these findings is somewhat sparse. I found the paper fascinating and certainly very impressive from a methodologic perspective. My concerns mostly involve the clarity of the methodologic and quantitative procedures, along with several conceptual concerns.

Critique: While the paper seems to focus on thalamic-cortical networks, the authors on several occasions refer to clarifying cortico-cortico networks, though these latter networks do not seem to be as well addressed by their data. I realize that they examine the impact of their subcortical input and recurrent strength measures on broad cortical intrinsic networks in GE and TLE, but there is some misplaced emphasis at times. The authors might consider limiting their focus to thalamo-cortico relations, thus carrying through on the implication in their data that the thalamus is the driver of cortical seizures abnormalities even in the case of "focal" TLE. Alternatively, they can organize their

results so that their conclusions about strictly thalmo-cortico versus cortical abnormalities (and cortico-cortico) effects are better demarcated and clearer.

The authors should make clear whether the relaxed mean field analysis model they present has been utilized before in the setting of epilepsy, and what advantages this technique brings relative to other methods that purport to link features of structural and functional connectomes.

The microcircuit measures involving recurrent excitation/inhibition (w) and subcortical drive (l) are not well explained in terms of how they are derived, and what assumptions they are using when their model produces these values (along with a coupling constant, G) when inputting the resting state and structural connectivity matrices. The conceptual role of these microcircuit measures is unclear, and they are not even mentioned in the introduction until the final sentence.

The explanation of the relaxed mean field theory model needs to be laid out in more user-friendly terms. That is, the mathematical explanation is of course needed, but a summary conveying how it captures the links and relations between structure and function should be more clearly and simply described.

It is unclear if the authors are using their data to argue for a particular causal chain. At times, they seem to imply that the microcircuit mechanisms explain macroscopic epilepsy expression, with the macroscopic features measured through volume and shape analysis. In this causal chain they are emphasizing thalamic-driven cortical effects. If they are arguing for this causal chain, it is not well articulated. More generally, they need to make clear to what degree their data addresses any particular causal chain, and they should note several alternative causal chains, as their data is not definitive in this regard. .

The investigators examined the role of drug response, history of secondary generalized tonic-clonic seizures, and the degree of hippocampal abnormality, and found that their results generally held up against these rival explanatory factors. The GE and TLE subgroups can have very different ages of onset and illness durations. I encourage them to report whether group differences hold up after accounting for these factors.

The results overall support the notion that both generalized epilepsy and temporal lobe epilepsy are associated with cortical-thalamic network abnormalities at multiple levels. Both disorders seem particularly sensitive to network alterations involving the limbic system. This limbic involvement is intuitive for TLE, but not so for GE. Why might GE share with TLE this preponderance for limbic localized effects – and does the thalamus play the same role in each disorder setting up such limbic effects?

I wonder about their use of the term microstructural. Their microstructural measure are ultimately reliant upon standard diffusion measures such as MD and resting state measures such as connection strength. The degree to which their excitation/inhibition and subcortical drive measures represent microstructure can be questioned as the use of a parcellation scheme for the inputted data (n.b., resting state and diffusion regions of interest) involves a fairly coarse and broad spatial resolution.

The authors discuss how the mechanisms of the functional connectome that they have measured act like a “filter” on the structural connectome. Their use of the concept of filtering is unclear. Are they referring to a selection process (if so, what are the unwanted parts that the filter removes); or does “filter” refer to something else -- a transformative process, a mediating process?

The authors report that the increased subcortical drive in generalized epilepsy did not markedly

perturb recurrent excitation/inhibition in cortical regions, a finding that mirrors the small degree of morphological and microstructural abnormalities observed. On the other hand, TLE showed reductions in subcortical drive in limbic and somatomotor areas, and this occurred in parallel with increased recurrent excitation/inhibition, echoing the stronger morphologic and microstructural abnormality detected in frontocentral cortices. The increased subcortical drive in GE without changes in increased excitation/inhibition is interpreted as consistent with animal evidence that recurrent loops between the thalamus and neocortical areas promote generalized seizure activity. If this key finding regarding GE (increased subcortical drive) does not modulate recurrent loops, how are their finding consistent the notion the recurrent loops promote generalize activity. Also, it is not clear why these data for GE involving increased subcortical drive with changes in recurrent strength necessarily limits its cortical impact. The impact of subcortical drive on cortex need not be limited to sustaining cortical loops.

The reduced subcortical drive and increased excitation/inhibition in TLE is interpreted to reflect a disconnection of limbic cortices, leading to functional segregation from large-scale intrinsic networks. This pattern is thought to set up recurrent cortico-cortical loops in TLE. But what exactly is the subcortical drive doing -- is it inhibitory to these loops? Couldn't these cortical loops reflect occult epileptogenesis in cortical regions apart from any subcortical influence?

Lastly, it was unclear how precisely the recurrent cortico-cortical loops might reflect or mirror the morphologic abnormalities, as the cortico-cortical loops are only described in terms of established intrinsic functional connectivity networks, networks which are known to be multi-regional throughout the brain.

Response to Reviewers (COMMSBIO-19-1598-T)

We thank the Reviewers for their positive evaluations and the Editors for giving us the opportunity to submit a revised manuscript. We are grateful to all Reviewers for their time and highly constructive comments. We found their suggestions very helpful for improving the quality of our paper. We have addressed all suggestions of the Editors and Reviewers in a point-by-point fashion and highlighted corresponding changes in the manuscript in yellow.

EDITOR

As you can see reviewer 2 asks for additional analysis taking into account ages of onset and illness durations. The other concerns raised by the reviewers largely relate to the presentation.

We thank the Editor for giving us the opportunity to submit a revised manuscript. We carried out the suggested additional analyses (all findings remained consistent) and improved the presentation of our work. Please see the detailed point-by-point responses to the individual Reviewers.

REVIEWER #1

The manuscript “Macroscale and microcircuit dissociation of focal and generalised human epilepsies” presented multi-scale analyses, such as multimodal imaging, connectomics and computational simulations to dissociate human focal (TLE) and generalised epilepsy (GE) on the thalami-cortical pathology, especially the prominent role of thalami-cortical loops. The authors find that TLE shows more severe microcircuit and macro scale imbalances, e.g., stronger cortical and thalamic atrophy, more bilateral volume reduction of thalamus, as well as widespread reductions in connectivity and reduced subcortical drive to the limbic and somatomotor networks. These new and solid evidences endow stronger support to dissociate human focal and generalised epilepsy at both the macroscale and microcircuit levels. The experimental findings and computational simulations are solid and significant, and the results are interesting and significant for both theoretical and experimental researchers in the field of epilepsy, especially for people who are trying to understand the underlying biophysical mechanisms of the seizure generalisation from the focal seizures.

We thank the Reviewer for recognizing the novelty, solidity, and significance of our findings and for the constructive comments.

However, I am still a little confused by the motivations and the discussions in the work have not touched this topic: The authors only presented results to dissociate TLE and GE, but did not try to clarify the non-consensus on the similarity or divergence on the role of thalami-cortical pathology in both generalised and focal epilepsies. As shown in the results of the manuscript, the difference between TLE and GE is so significant, even considering the comparison between TLE with secondary generalisation and GE. I am puzzled why there was nonconsensus and whether and how this work can clarify the nonconsensus. If so, the work can evoke more discussions on this problem and inspire further study. In my view, this part should be clearly introduced, presented and discussed in details. In this way, this manuscript will be more instrumental to understand mechanisms of focal and generalised human epilepsies, and thus contribute much more to this field.

We thank the Reviewer for this remark and agree the formulation on limited consensus was imprecise. In response, we have elaborated on the study justification in *Abstract (P.2)*, *Introduction (P.3)*, and *Discussion (P.10)*.

“Thalamo-cortical pathology plays key roles in both generalized and focal epilepsies, but there is little work directly comparing these syndromes at the level of whole-brain mechanisms.”

“Importantly, there are no studies directly comparing focal and generalized epilepsies in terms of thalamo-cortical, cortico-cortical network pathology, and underlying functional dynamics. Furthermore, neuroimaging findings have

thus far not been related to potential microcircuit mechanisms that play a critical role in shaping the macroscopic expression of epilepsy-related network abnormalities¹.”

“Yet, differential patterns of cortico-cortical and thalamo-cortical compromise across the spectrum of human epilepsies are not fully understood, due to the scarcity of work systematically and directly comparing focal and generalized epilepsies.”

Besides, the manuscript needs careful re-editing to clear the typos, grammatical mistakes as well as to carefully arrange the positions of the abbreviations.

We thank the Reviewer for pointing this out and have carefully the revised manuscript.

REVIEWER #2

This paper presents an interesting connectivity study that compares the brains of people with generalized and focal epilepsies and compares these with controls. The study is well thought out, structured and reported. It presents novel results that could impact upon future understanding of these conditions and their treatment. Each aspect of the study provides useful and different insights that come together into a coherent story.

We thank the Reviewer for the positive evaluation and appreciate the constructive suggestions.

One of the key results is that there are “more marked anomalies in TLE relative to controls and GE”. However, it is not clear how much this might related to lesions in the cortices of those with TLE. The selection criteria were no mass lesion and hippocampal lesions were considered, could individual’s morphological and connectivity profiles be influenced by not-so-major cortical lesions that would come up as a broader effect in the combined results?

This is an interesting suggestion, which we have addressed in the revised *Discussion* (P.11).

“Our inclusion criteria ruled out the presence of visible neocortical lesions in the investigated patient cohorts and, overall, the pathological substrates of grey matter atrophy in TLE remain unclear. Yet, cortical atrophy appears relatively prevalent across TLE cohorts, with marked gray matter loss of temporal and fronto-central regions seen in approximately 35% of patients². Although the presence of subtle pre-existing anomalies of neurodevelopmental origin is possible³, cortical atrophy undergoes measurable progressive changes over time^{4,7}. These finding are also in line with the conclusions of a prior post-mortem study in TLE patients, suggesting acquired cortical pathology that potentially represents seizure related damage and manifests as both gliotic and microgliotic changes⁸. In the thalamus, while post-mortem analysis also emphasized heterogeneity across patients, findings overall confirmed reductions in neuronal density in mediodorsal divisions, which could represent a plausible pathological substrate of the local volume reductions seen in the current work⁹.”

I am concerned about some of the details of the patients as presented in Table 1, and what impact the extremes might have upon the results. For instance, some patients have durations of 0.01 and 0.08 years, and some have seizure frequencies of 0.01 and 0.03 per month. These seem rather extremely small and further justification of inclusion of these patients is needed.

As suggested, we provided further details on the included patient cohort. We also carried out an additional sensitivity analysis that removed patients with durations shorter than 1 year and those with seizure frequencies of less than 1/year. Results are highly similar to our main findings. These analyses are now presented in the new **Supplementary Figure 7**. Details are provided on P. 9

“Furthermore, findings were robust when restricting our cohorts to patients with at least 1-year epilepsy duration and at least 1 seizure/year (94 TLE, 74 GE) (SUPPLEMENTARY FIGURE 7).”

The model description in the biophysical simulations would benefit from increased explanations of the

different components of the model for those who are less familiar with the models. Brief explanations of the forms of equations 1-3 would be useful. The model is stated to implement a “spiking neuronal network” but there is no spiking mechanism but rather a spiking rate function – an explanation of this would be helpful.

We thank the Reviewer for this helpful remark. We included additional details on the model in the revised *Methods* (P.17) and further clarified the approximation of the spiking model behavior.

“For further details on the model and its mathematical underpinnings, we refer to the original publication on the relaxed mean field model¹⁰ and earlier work on the use of (non-relaxed) mean field models in the context of connectomics¹¹.

In brief, mean-field neural mass models capture neural dynamics at the level of neuronal populations. This is achieved by the mathematical simplification (i.e., mean field reduction) of detailed spiking neuronal networks. Mean field models have been shown to capture complex neural dynamics with relatively low parametric complexity¹². More specifically, under the mean field model, the neural dynamics of a given region are governed by four components: (i) recurrent intraregional input, i.e., recurrent excitation-inhibition; (ii) inter-regional input, mediated by anatomical connections from other nodes, (iii) extrinsic input, mainly from subcortical regions, and (iv) neuronal noise¹⁰. There are “free” parameters associated with each component. The original (non-relaxed) mean field models¹¹ assume these parameters are constant across brain regions, which is not biologically plausible. The relaxed mean field model that we utilized allows recurrent excitation-inhibition and extrinsic (subcortical) input to vary across regions and automatically estimated¹⁰. The dynamics of each region is described by the following coupled nonlinear stochastic differential equations¹¹:

$$\dot{S}_i = -\frac{S_i}{\tau_s} + r(1 - S_i)H(x_i) + \sigma v_i(t) \quad (1)$$

$$H(x_i) = \frac{ax_i - b}{1 - \exp(-d(ax_i - b))} \quad (2)$$

$$x_i = wJS_i + GJ \sum_j C_{ij}S_j + I \quad (3)$$

For a given region i , S_i in formula (1) represents the average synaptic gating variable, $H(x_i)$ in formula (2) denotes the population firing rate, and x_i in formula (3) denotes the total input current.”

Minor:

- In the first sentence of the Introduction, clarify what “flexible” means.

We thank the Reviewer for this observation. We also felt that the qualifier was redundant, and have removed it from the revised sentence (P. 3).

“Widespread recurrent connections between thalamus and cortex represent a cardinal principle of brain organization, and thalamic-cortical interplay has long been recognized to contribute to whole-brain function and dysfunction in both health and disease.”

- The figures contain a lot of text that is really small and difficult to read on the printed page. I suggest increasing the font size of these smallest fonts.

We followed the Reviewer’s suggestion, and adjusted the font sizes on the figures.

- I would like to know that the assumptions of the statistical tests were correctly considered.

The current study benefitted from a relatively large sample size and surface-based MRI feature data underwent diffusion kernel smoothing; both factors are known to increase data normality and the assumptions of parametric tests. These tests are also in keeping with previous studies from our group and others in healthy and diseased populations (e.g., Bernhardt et al. 2013, Neurology; Valk et al. 2016 Cerebral Cortex, Makowski et al. 2016 NjP Schizophrenia, Ecker et al. 2013 PNAS). Tests were implemented using SurfStat toolbox (used in >300 publications) co-developed by members of the study team. We furthermore verified data skewness across all cortical surface vertices for morphological and microstructural measures. For cortical thickness, skewness was 0.01 ± 0.33 across vertices; for FA, it was 0.86 ± 0.15 ; for MD, it was 0.77 ± 1.62 . Overall, these findings indicate that normality assumptions were

met. Of note, main findings on the dissociation between GE and TLE were relatively consistent when data were log transformed (which stabilizes potential deviations in variance).

- The proliferation of abbreviations is a little tiresome; I suggest keeping the abbreviations to only those terms used most often.

We thank the Reviewer for the suggestion and removed the most seldomly used abbreviations from the manuscript, namely: MFM, rMFM rs-fMRI, sGTCS, SWM, SPHARM, DWI, BOLD, and FOD.

REVIEWER #3:

Summary: The chief novelty of this study involves utilizing a relaxed mean field model as a computational technique to estimate linkages between structural and functional connectome measures, namely micro-circuit parameters produced from resting state and diffusion, along with macro-circuit features from brain volumetrics and surface shape analyses. The specific microcircuit parameters, whose calculation is not described, involve recurrent excitation/inhibition, in addition to external subcortical drive. These allow the authors to generate veridical simulations of functional connectivity from a structural connectome (diffusion based) that can be related linked to intrinsic functional connectivity networks and macrostructural cortical and thalamic morphology features of shape and volume. They contrast generalized epilepsy (GE) and temporal lobe epilepsy (TLE), with a focus on thalamo-cortical relationships and connectivity. The results clearly showed that the putatively more focal disorder of TLE is associated with widespread atrophy and surface-shaped changes in gray and white matter involving the thalamus and broader brain regions, a somewhat surprising finding given the broader electrophysiologic brain effects of GE. The relaxed mean field model technique allowed the investigators to examine measures related to subcortical drive and recurrent excitation/inhibition of cortico-cortico dynamics, as well as thalamic-cortical dynamics, and, again, TLE displayed more abnormalities than GE and in comparison to normal controls. More specifically, compared to GE, TLE showed decreases in subcortical drive that appeared related to increases in excitation/inhibition, particularly in limbic and fronto-central networks. In turn, this pattern in TLE demonstrated a greater association with multiple cortical intrinsic networks. In contrast, GE presented with increased subcortical drive, but no change in recurrent cortical excitation inhibition. The interpretation of these findings is somewhat sparse. I found the paper fascinating and certainly very impressive from a methodologic perspective. My concerns mostly involve the clarity of the methodologic and quantitative procedures, along with several conceptual concerns.

We thank the Reviewer for finding the paper fascinating and impressive and are thankful for the thoughtful and constructive suggestions.

Critique: While the paper seems to focus on thalamic-cortical networks, the authors on several occasions refer to clarifying cortico-cortico networks, though these latter networks do not seem to be as well addressed by their data. I realize that they examine the impact of their subcortical input and recurrent strength measures on broad cortical intrinsic networks in GE and TLE, but there is some misplaced emphasis at times. The authors might consider limiting their focus to thalamo-cortico relations, thus carrying through on the implication in their data that the thalamus is the driver of cortical seizures abnormalities even in the case of “focal” TLE. Alternatively, they can organize their results so that their conclusions about strictly thalamo-cortico versus cortical abnormalities (and cortico-cortico) effects are better demarcated and clearer.

We followed the second alternative suggested by the Reviewer and reorganized several passages, including the *Abstract* (P. 2):

“...we examined cortico-cortical and thalamo-cortical signatures...”

We also slightly reorganized the *Results* section to better highlight whether we are referring to cortical (and cortico-cortical) or thalamic (and thalamo-cortical) findings. See P.4

“The main findings show direct comparisons between patient cohorts for cortical (and cortico-cortical) as well as thalamic (and thalamo-cortical) networks, at the level of morphology, microstructure, connectivity and microcircuit organization.”

To improve consistency across the *Results* subsections, we always describe the cortical findings first, followed by the thalamic findings. As such, we slightly reorganized the *Connectivity and microcircuit simulations* section, as well as **Figure 3** and **Supplementary Figure 2**.

“We first built connectome-informed mean field models to estimate the contributions of subcortical drive and recurrent excitation-inhibition on simulated cortico-cortical dynamics, and then targeted thalamo-cortical functional networks using a seed-based resting-state functional connectivity analysis.”

The authors should make clear whether the relaxed mean field analysis model they present has been utilized before in the setting of epilepsy, and what advantages this technique brings relative to other methods that purport to link features of structural and functional connectomes. To our knowledge, this method has not yet been applied to epilepsy, which we now outline in the revised *Introduction* and *Discussion*. We also comment on increased biophysical interpretability of these techniques compared to statistical approaches such as ICA. *See P. 3-4*

“Moreover, the synergistic integration of these different modalities addresses structure-function coupling and can be harnessed to provide large-scale simulations of brain function^{10,13-15}. One approach specifically models whole-brain dynamics via a network of anatomically connected neural masses. In contrast to statistical approaches that interrogate macroscale organization and structure-function coupling, the dynamical properties in these models are governed by parameters with biophysically plausible interpretations that reflect empirical models of neuronal circuit function¹⁰. A recent study in healthy individuals has also shown that these models can make robust simulations of functional connectivity from structural connectivity, and that the model can furthermore be used to infer regional-specific microcircuit parameters, such as recurrent excitation-inhibition and external subcortical drive¹⁰. Thus, applying these novel models to epilepsy will provide a complementary and mechanistic perspective on the role of the thalamo-cortical system on cortico-cortical dynamics.”

And P. 12

“Capitalizing on the first application of these relaxed mean field models to epilepsy cohorts, we identified diverging mechanisms in focal and generalized epilepsies, bridging microcircuit properties and macroscale function.”

The microcircuit measures involving recurrent excitation/inhibition (w) and subcortical drive (l) are not well explained in terms of how they are derived, and what assumptions they are using when their model produces these values (along with a coupling constant, G) when inputting the resting state and structural connectivity matrices. The conceptual role of these microcircuit measures is unclear, and they are not even mentioned in the introduction until the final sentence.

We thank the Reviewer for this comment. As recommended, we expanded our descriptions of the model in the revised *Introduction* (see also previous answer) and *Results*. *P. 7*.

“For biophysical circuit simulations, we leveraged a relaxed mean field approach that models whole brain functional dynamics and connectivity based on diffusion MRI connectome information (see Methods for details). In brief, these models assume that neural dynamics are governed by (i) recurrent intraregional input related to recurrent excitation-inhibition, (ii) inter-regional input mediated by anatomical connections from other nodes, (iii) extrinsic input mainly mediated by subcortical regions, and (iv) neuronal noise¹⁰. While the originally proposed mean field models assume these four parameters to be constant across brain regions, the relaxed mean field model applied here allows recurrent excitation-inhibition and subcortical input to vary, and to estimate these parameters at a region-specific level¹⁰.”

We also updated the *Methods* section to provide more in-depth and conceptual details. *See P. 17*.

“For further details on the model and its mathematical underpinnings, we refer to the original publication on the relaxed mean field model ¹⁰ and earlier work on the use of (non-relaxed) mean field models in the context of connectomics ¹¹.”

In brief, mean-field neural mass models capture neural dynamics at the level of neuronal populations. This is achieved by the mathematical simplification (i.e., mean field reduction) of detailed spiking neuronal networks. Mean field models have been shown to capture complex neural dynamics with relatively low parametric complexity ¹². More specifically, under the mean field model, the neural dynamics of a given region are governed by four components: (i) recurrent intraregional input, i.e., recurrent excitation-inhibition; (ii) inter-regional input, mediated by anatomical connections from other nodes, (iii) extrinsic input, mainly from subcortical regions, and (iv) neuronal noise ¹⁰. There are “free” parameters associated with each component. The original (non-relaxed) mean field models ¹¹ assume these parameters are constant across brain regions, which is not biologically plausible. The relaxed mean field model that we utilized allows recurrent excitation-inhibition and extrinsic (subcortical) input to vary across regions and automatically estimated ¹⁰. The dynamics of each region is described by the following coupled nonlinear stochastic differential equations ¹¹:

$$\dot{S}_i = -\frac{S_i}{\tau_S} + r(1 - S_i)H(x_i) + \sigma v_i(t) \quad (1)$$

$$H(x_i) = \frac{\alpha x_i - b}{1 - \exp(-d(\alpha x_i - b))} \quad (2)$$

$$x_i = wJ S_i + GJ \sum_j C_{ij} S_j + I \quad (3)$$

For a given region i , S_i in formula (1) represents the average synaptic gating variable, $H(x_i)$ in formula (2) denotes the population firing rate, and x_i in formula (3) denotes the total input current.”

The explanation of the relaxed mean field theory model needs to be laid out in more user-friendly terms. That is, the mathematical explanation is of course needed, but a summary conveying how it captures the links and relations between structure and function should be more clearly and simply described.

We thank the Reviewer for this remark and elaborated the description of the model in lay terms in both the *Introduction (P.4)*

“One approach specifically models whole-brain dynamics via a network of anatomically connected neural masses. In contrast to statistical approaches that interrogate macroscale organization and structure-function coupling, the dynamical properties in these models are governed by microcircuit parameters with biophysical interpretations that reflect empirical models of neuronal circuit function ¹⁰. A recent study in healthy individuals has also shown that these models can make robust simulations of functional connectivity from structural connectivity, and that the model can furthermore be used to infer regional-specific microcircuit parameters, such as recurrent excitation-inhibition and external subcortical drive ¹⁰. Thus, applying these novel models to epilepsy will provide a complementary and mechanistic perspective on the role of the thalamo-cortical system on cortico-cortical dynamics.”

And the revised *Methods* (see previous comment).

It is unclear if the authors are using their data to argue for a particular causal chain. At times, they seem to imply that the microcircuit mechanisms explain macroscopic epilepsy expression, with the macroscopic features measured through volume and shape analysis. In this causal chain they are emphasizing thalamic-driven cortical effects. If they are arguing for this causal chain, it is not well articulated. More generally, they need to make clear to what degree their data addresses any particular causal chain, and they should note several alternative causal chains, as their data is not definitive in this regard.

We thank the reviewer for this very important suggestion. We further elaborated on the potential causal effects that may contribute to these findings and how these can be further investigated. See *P. 14*.

“We conclude by emphasizing that our work offers an integrative perspective on how different scales of brain organization interact in focal and generalized epilepsies. While our cross-sectional findings preclude direct inferences on whether microcircuit perturbations cause macroscale reorganization (or vice versa), they motivate future longitudinal research that models these multi-scale interactions in the context of disease progression. Given their scope to interrogate microcircuit and macroscale effects, we will similarly explore the utility of the developed platform in the prediction and monitoring of the efficacy of therapeutic interventions, notably anti-epileptic medication and epilepsy surgery.”

The investigators examined the role of drug response, history of secondary generalized tonic-clonic seizures, and the degree of hippocampal abnormality, and found that their results generally held up against these rival explanatory factors. The GE and TLE subgroups can have very different ages of onset and illness durations. I encourage them to report whether group differences hold up after accounting for these factors.

We thank the Reviewer for this important suggestion. We carried out the additional sensitivity analysis after controlling for age of onset and duration. Findings remained overall consistent and this now reported in the revised *Results* section (P. 9)

“As both measures can similarly affect brain structure and function, we furthermore ran a model that additionally controlled for age at onset and duration.”

further highlighted in the *Discussion* P.14

“A comprehensive battery of sensitivity analyses revealed that between-cohort divergences were only modestly related to potential variations in drug-response patterns, virtually identical when restricting the TLE cohort to those with a history of secondary generalized tonic-clonic seizures, and robust against correction for age at seizure onset and disease duration.”

And in the new **SUPPLEMENTARY FIGURE 6.**

The results overall support the notion that both generalized epilepsy and temporal lobe epilepsy are associated with cortical-thalamic network abnormalities at multiple levels. Both disorders seem particularly sensitive to network alterations involving the limbic system. This limbic involvement is intuitive for TLE, but not so for GE. Why might GE share with TLE this preponderance for limbic localized effects – and does the thalamus play the same role in each disorder setting up such limbic effects?

This is an excellent suggestion. We further discussed the increased susceptibility of the limbic system on P. 13.

“Our results support the notion that both generalized epilepsy and temporal lobe epilepsy are associated with thalamo-cortical and cortico-cortical network abnormalities at multiple levels. Moreover, our findings demonstrated a robust dissociation between TLE and GE, with the former being more markedly affected. Despite this divergence, one cortical network that appeared most susceptible to epilepsy-related pathology overall was the limbic system, showing measurable microstructural anomalies in both focal and generalized syndromes. Increased susceptibility of the limbic system has repeatedly been suggested in the context of TLE^{16,17}, and is also plausible given the mesiotemporal epicenter of the disorder. In different GE syndromes, several studies have also detected structural anomalies in limbic areas, including work showing mesiotemporal gray matter loss as well as malrotation¹⁸⁻²⁰. The increased disease effect on the limbic system across both TLE and GE may relate to the role of the thalamus itself in the pathophysiological networks of both syndromes. Several divisions of the thalamus, such as the anterior and mediodorsal division, have traditionally been grouped to the limbic circuitry due to their anatomical proximity and dense connectivity with several key nodes of the “grand lobe limbique,” such as the hippocampus and amygdala. On the other hand, limbic and paralimbic cortices differ from other cortices in terms of their underlying microstructure and circuit properties^{21,22}, and have a relatively simple laminar organization compared to eulaminate areas exhibiting a more differentiated lamination^{21,23,24}. These differential patterns also mirror gradual

changes in myeloarchitecture and microcircuit properties, with limbic areas showing less myelination but higher synaptic density compared to areas involved in less integrative sensory-motor and unimodal association processing. The latter are ultimately believed to affect the increased plasticity and heightened disease susceptibility of the limbic network to multiple diseases²¹.”

I wonder about their use of the term microstructural. Their microstructural measure are ultimately reliant upon standard diffusion measures such as MD and resting state measures such as connection strength. The degree to which their excitation/inhibition and subcortical drive measures represent microstructure can be questioned as the use of a parcellation scheme for the inputted data (n.b., resting state and diffusion regions of interest) involves a fairly coarse and broad spatial resolution.

We thank the Reviewer for this remark. To clarify, the term ‘microstructural’ was exclusively used in the context of the diffusion MRI findings and not for the analysis of resting-state data. Of note, these findings were derived via surface-based analyses in cortical areas, i.e. not based on a parcellation, and in the thalamus. We added additional clarifications at several locations in the manuscript. Furthermore, we provided additional context on the microstructural interpretations of these findings. (P. 12).

“As the diffusion parameter changes identified in this work may be sensitive to different microstructural and architectural substrates, we can only speculate on histopathological underpinnings. In TLE, prior work has shown that fractional anisotropy reductions of select fiber tracts may relate to changes in axonal membrane and myelination²⁵, whereas mean diffusivity changes may represent a combination of gliosis and increased extracellular space^{26,27}.”

The authors discuss how the mechanisms of the functional connectome that they have measured act like a “filter” on the structural connectome. Their use of the concept of filtering is unclear. Are they referring to a selection process (if so, what are the unwanted parts that the filter removes); or does “filter” refer to something else -- a transformative process, a mediating process?

We apologize for the unclear formulation, removed the filter analogy, and rephrased the respective passage on P. 12.

“The relaxed mean field models chosen here provide plausible estimates of functional connectivity from structural connectomes with relatively few assumptions and low parametric complexity.”

The authors report that the increased subcortical drive in generalized epilepsy did not markedly perturb recurrent excitation/inhibition in cortical regions, a finding that mirrors the small degree of morphological and microstructural abnormalities observed. On the other hand, TLE showed reductions in subcortical drive in limbic and somatomotor areas, and this occurred in parallel with increased recurrent excitation/inhibition, echoing the stronger morphologic and microstructural abnormality detected in frontocentral cortices. The increased subcortical drive in GE without changes in increased excitation/inhibition is interpreted as consistent with animal evidence that recurrent loops between the thalamus and neocortical areas promote generalized seizure activity. If this key finding regarding GE (increased subcortical drive) does not modulate recurrent loops, how are their finding consistent the notion the recurrent loops promote generalize activity. Also, it is not clear why these data for GE involving increased subcortical drive with changes in recurrent strength necessarily limits its cortical impact. The impact of subcortical drive on cortex need not be limited to sustaining cortical loops.

We thank the Reviewer for this comment. We further elaborated on our argument on P. 12.

“The increased subcortical drive in GE did not seem to be accompanied by marked changes in recurrent excitation-inhibition in cortical regions, a finding that may mirror the overall smaller degree of morphological and microstructural anomalies observed in our analyses. Thus, despite unchanged intraregional recurrent input, cortical circuits in GE may thus still show increased excitability due to increased subcortical drive.”

The reduced subcortical drive and increased excitation/inhibition in TLE is interpreted to reflect a disconnection of limbic cortices, leading to functional segregation from large-scale intrinsic networks. This pattern is thought to set up recurrent cortico-cortical loops in TLE. But what exactly is the subcortical drive doing -- is it inhibitory to these loops? Couldn't these cortical loops reflect occult epileptogenesis in cortical regions apart from any subcortical influence?

We thank the Reviewer for the thoughtful comment. To clarify, in the relaxed mean field model, the subcortical input parameter is modelled as being excitatory, so that reduced drive would indeed be interpreted as resulting in less excitation. The Reviewer's suggestion on potential implications for cortical epileptogenesis are in line with our initial argument, which we hopefully clarified in the revised *Discussion*. In particular, we removed words implying a causal link between reduced subcortical input and the formation of cortico-cortical loops in TLE. See *P. 12*.

“The overall disconnection of limbic cortices may possibly relate to a functional segregation from subcortical networks, which may also reflect the existence of recurrent cortico-cortical loops in TLE, and potentially promote epileptogenesis.”

Lastly, it was unclear how precisely the recurrent cortico-cortical loops might reflect or mirror the morphologic abnormalities, as the cortico-cortical loops are only described in terms of established intrinsic functional connectivity networks, networks which are known to be multi-regional throughout the brain.

The Reviewer is correct that the morphological measures are region-specific. With respect to the recurrent loop interpretation, we made it more explicit that this was made based on region-specific microcircuit estimates as well, obtained from the relaxed mean field models. Thus, while the MFMs take multi-regional structural and functional connectivity information as inputs, they provide regional estimates of recurrent excitation-inhibition. Please see the revised *Discussion P. 12*.

“In fact, these large-scale models allowed for the estimation of region-specific model parameters that provided insights into the role of excitation-inhibition and excitatory subcortical drive on microcircuit function. [...] The increased subcortical drive in GE did not seem to be accompanied by marked changes in recurrent excitation-inhibition in cortical regions, a finding that may mirror the overall smaller degree of morphological and microstructural anomalies observed in our analyses. Despite unchanged intraregional recurrent input, cortical circuits in GE may still show greater excitability due to increased subcortical drive.”

REFERENCES FOR RESPONSE LETTER

- 1 Farrell, J. S., Nguyen, Q. A. & Soltesz, I. Resolving the Micro-Macro Disconnect to Address Core Features of Seizure Networks. *Neuron* **101**, 1016-1028, doi:10.1016/j.neuron.2019.01.043 (2019).
- 2 Bernhardt, B. C. *et al.* Mapping limbic network organization in temporal lobe epilepsy using morphometric correlations: insights on the relation between mesiotemporal connectivity and cortical atrophy. *NeuroImage* **42**, 515-524, doi:10.1016/j.neuroimage.2008.04.261 (2008).
- 3 Voets, N. L., Bernhardt, B. C., Kim, H., Yoon, U. & Bernasconi, N. Increased temporolimbic cortical folding complexity in temporal lobe epilepsy. *Neurology* **76**, 138-144, doi:10.1212/WNL.0b013e318205d521 (2011).
- 4 Bernhardt, B. C., Bernasconi, N., Concha, L. & Bernasconi, A. Cortical thickness analysis in temporal lobe epilepsy: reproducibility and relation to outcome. *Neurology* **74**, 1776-1784, doi:10.1212/WNL.0b013e3181e0f80a (2010).
- 5 Bernhardt, B. C. *et al.* Longitudinal and cross-sectional analysis of atrophy in pharmaco-resistant temporal lobe epilepsy. *Neurology* **72**, 1747-1754, doi:10.1212/01.wnl.0000345969.57574.f5 (2009).
- 6 Galovic, M. *et al.* Progressive Cortical Thinning in Patients With Focal Epilepsy. *JAMA Neurol*, doi:10.1001/jamaneurol.2019.1708 (2019).
- 7 Caciagli, L. *et al.* A meta-analysis on progressive atrophy in intractable temporal lobe epilepsy *Neurology* **89**(5):506-516 (2017).
- 8 Blanc, F. *et al.* Investigation of widespread neocortical pathology associated with hippocampal sclerosis in epilepsy: a postmortem study. *Epilepsia* **52**, 10-21, doi:10.1111/j.1528-1167.2010.02773.x (2011).
- 9 Sinjab, B., Martinian, L., Sisodiya, S. M. & Thom, M. Regional thalamic neuropathology in patients with hippocampal sclerosis and epilepsy: A postmortem study. *Epilepsia* **54**, doi:10.1111/epi.12403 (2013).
- 10 Wang, P. *et al.* Inversion of a large-scale circuit model reveals a cortical hierarchy in the dynamic resting human brain. *Science Advances* **5**, eaat7854, doi:10.1126/sciadv.aat7854 (2019).
- 11 Deco, G. *et al.* Resting-state functional connectivity emerges from structurally and dynamically shaped slow linear fluctuations. *The Journal of neuroscience : the official journal of the Society for Neuroscience* **33**, 11239-11252, doi:10.1523/JNEUROSCI.1091-13.2013 (2013).
- 12 Deco, G., Jirsa, V. K. & McIntosh, A. R. Emerging concepts for the dynamical organization of resting-state activity in the brain. *Nature reviews. Neuroscience* **12**, 43-56, doi:10.1038/nrn2961 (2011).
- 13 Schirner, M., McIntosh, A. R., Jirsa, V., Deco, G. & Ritter, P. Inferring multi-scale neural mechanisms with brain network modelling. *eLife* **7**, doi:10.7554/eLife.28927 (2018).
- 14 Cabral, J., Kringelbach, M. L. & Deco, G. Functional connectivity dynamically evolves on multiple time-scales over a static structural connectome: Models and mechanisms. *NeuroImage* **160**, 84-96, doi:10.1016/j.neuroimage.2017.03.045 (2017).
- 15 Jirsa, V. K., Sporns, O., Breakspear, M., Deco, G. & McIntosh, A. R. Towards the virtual brain: network modeling of the intact and the damaged brain. *Archives italiennes de biologie* **148**, 189-205 (2010).
- 16 Adler, S. *et al.* Topographic principles of cortical fluid-attenuated inversion recovery signal in temporal lobe epilepsy. *Epilepsia* **59**, 627-635, doi:10.1111/epi.14017 (2018).
- 17 Bernhardt, B. C. *et al.* Preferential susceptibility of limbic cortices to microstructural damage in temporal lobe epilepsy: A quantitative T1 mapping study. *NeuroImage*, doi:10.1016/j.neuroimage.2017.06.002 (2017).
- 18 Caciagli, L. *et al.* Abnormal hippocampal structure and function in juvenile myoclonic epilepsy and unaffected siblings. *Brain : a journal of neurology* **142**, 2670-2687, doi:10.1093/brain/awz215 (2019).
- 19 Zhou, S. Y. *et al.* Selective medial temporal volume reduction in the hippocampus of patients with idiopathic generalized tonic-clonic seizures. *Epilepsy research* **110**, 39-48, doi:10.1016/j.eplepsyres.2014.11.014 (2015).
- 20 Tondelli, M., Vaudano, A. E., Ruggieri, A. & Meletti, S. Cortical and subcortical brain alterations in Juvenile Absence Epilepsy. *NeuroImage. Clinical* **12**, 306-311, doi:10.1016/j.nicl.2016.07.007 (2016).
- 21 Garcia-Cabezas, M. A., Zikopoulos, B. & Barbas, H. The Structural Model: a theory linking connections, plasticity, pathology, development and evolution of the cerebral cortex. *Brain structure & function* **224**, 985-1008, doi:10.1007/s00429-019-01841-9 (2019).
- 22 Goulas, A., Margulies, D. S., Bezgin, G. & Hilgetag, C. C. The architecture of mammalian cortical connectomes in light of the theory of the dual origin of the cerebral cortex. *Cortex; a journal devoted to the study of the nervous system and behavior*, doi:10.1016/j.cortex.2019.03.002 (2019).
- 23 Paquola, C. *et al.* Microstructural and functional gradients are increasingly dissociated in transmodal cortices. *PLoS biology* **17**, e3000284, doi:10.1371/journal.pbio.3000284 (2019).
- 24 Paquola, C. *et al.* Shifts in myeloarchitecture characterise adolescent development of cortical gradients. *eLife*, doi:<https://www.biorxiv.org/content/10.1101/706341v1.abstract> (2019).
- 25 Concha, L., Livy, D. J., Beaulieu, C., Wheatley, B. M. & Gross, D. W. In vivo diffusion tensor imaging and histopathology of the fimbria-fornix in temporal lobe epilepsy. *The Journal of neuroscience : the official journal of the Society for Neuroscience* **30**, 996-1002, doi:10.1523/JNEUROSCI.3033-09.2009 [pii] (2010).
- 26 Bernhardt, B. C. *et al.* The spectrum of structural and functional imaging abnormalities in temporal lobe epilepsy. *Annals of neurology* **80**, 142-153, doi:10.1002/ana.24691 (2016).
- 27 Wiesmann, U. C. *et al.* Water diffusion in the human hippocampus in epilepsy. *Magnetic resonance imaging* **17**, 29-36, doi:10.1016/0896-6275(99)00153-2 [pii] (1999).

REVIEWERS' COMMENTS:

Reviewer #3 (Remarks to the Author):

The authors responded well to my concerns and the issues I raised.

I have two smaller remaining concerns or suggestions.

1) The authors description of the mean field model is improved. They now make clear that the neural dynamics of a given region are governed by four components: recurrent intraregional input, inter-regional input, extrinsic input, and neuronal noise. They should now more clearly match up for the reader the terms in the MFM formula that match up or reflect these components, mentioning in the process an assumptions made by the formula.

2) The authors now note more clearly that their results were unchanged for those TLE patients with a history of secondary generalized seizures. They need to also make clear how they dealt with TLE patients who may have had even a small number of separate generalized seizure events in their history. I gather they were excluded.

We thank Reviewer 3 for the final remarks and have addressed these in this version. Specifically, we added 1) a sentence to clarify how we dealt with patients who may have had generalized seizures earlier in their history, 2) further descriptions of the MFM formula to enhance clarity.